# VR Games in Cultural Heritage: A Systematic Review of the Emerging Fields of Virtual Reality and Culture Games

Anastasios Theodoropoulos [1,*] and Angeliki Antoniou [2]

1   Department of Performing and Digital Arts, University of Peloponnese, 211 00 Nafplion, Greece
2   Department of Archival, Library and Information Studies, University of West Attica, 122 43 Egaleo, Greece
*   Correspondence: ttheodor@uop.gr

**Abstract:** In recent years, the use of VR games in cultural heritage has been growing. VR Games have increasingly found their way into museums and exhibitions, highlighting the increasing cultural value associated with games and the institutionalization of game culture. In particular, serious VR games have a variety of benefits for educational purposes. There are several studies that deployed VR games to improve visitor experiences in several contexts. However, there are not sufficient studies in the field that examine the benefits and drawbacks of VR gaming. This lack of classification studies is regarded as an obstacle to developing more effective games and proposing guidance on the best way of using them in cultural heritage. This review aims to analyze how VR games are used in cultural heritage settings, to explore the evolution and opportunities of this emerging field, the challenges and tensions these innovations present, and to collectively advance this work to benefit visitor experiences.

**Keywords:** VR games; culture games; heritage games; cultural heritage; virtual heritage; learning experiences with VR games

## 1. Introduction

The emergence of Virtual Reality (VR) has posed significant challenges for scientists who started investigating the way VR changes social dynamics [1] as well as social attitudes [2]. Especially during the pandemic, cultural heritage, culture, and tourism had to find new ways to engage the public, since travelling was largely restricted. Thus, the potential of virtual tourism was explored, showing multiple positive effects in promoting sites, allowing the recovery of tourist destinations after the pandemic, and supporting engagement with cultural content for remote audiences. [3]. In addition, virtual applications also allowed interactions with intangible cultural heritage [4]. When VR is combined with storytelling, e.g., 360 video storytelling, cultural content becomes engaging and significant increases in immersion and levels of presence are observed [5]. Finally, research has shown that VR places specific demands on human cognition, and in order to be engaging and provide a smooth experience, its design needs to respect human cognitive requirements [6].

In recent years, games in cultural heritage have been considered a good way to increase engagement with cultural content; thus, serious games and game-based learning is on the rise. Although challenges remain in their development, there are significant results showing advantages of their use [7,8]. Moreover, the combination of VR and games in cultural heritage also produced very promising results, especially for specific age groups, regardless of their domain knowledge of cultural heritage [9], making such games important tools for cultural experiences. Therefore, games and gamification processes in cultural heritage can not only increase engagement but also reflection and interpretation, and numerous studies seem to support this finding [10]. Previous reviews on the topic studied both Augmented Reality (AR) and VR games in cultural heritage, as an attempt to map the field and boost future research, focusing on three main aspects: sociability, gamification, and virtualization [11].

Considering the above, this study explores different ways to effectively incorporate VR games in cultural heritage through a systematic literature review (SLR). The analysis seeks to examine works on VR games, especially with empirical interventions with respect to immersion, motivation, engagement, and culture. Thankfully, there have been many recent studies incorporating VR games within culture settings.

The remainder of this article is organized as follows. Firstly, the background work presents a basic overview of the field of VR games in cultural heritage. Next, the methodology explains how the data for the current study were gathered. The results and discussion sections include both quantitative and qualitative findings. Finally, the study makes suggestions for future research based on the gaps in the literature.

## 2. Background Work

This section outlines basic elements of learning activities with games in cultural heritage, VR as a medium and presents the learning potential of VR games in the field as well as examples of relevant games.

### 2.1. Learning Experiences with Games in Cultural Heritage

Digital games or video games are games that integrate digital technology [12]. In videogames users play through audiovisual platforms with contents on the basis of a story created from historical or fantasy themes. The concept of digital games as represented in literature [12], Ref. [13] focuses on game itself, narrative (story), interactivity, and play. Games designed for educational objectives (serious games), look as an effective tool to learn cultural content in an engaging way. In order to transform static content into a serious game in Cultural Heritage, gamification techniques are used. Gamification is the process of introducing game strategies and components into some scenarios and situations that are not a game [14].

Additionally, there is a wide variety of cultural material since there is the tangible Cultural Heritage, which includes historic places and buildings, monuments, records, works of art, machineries, and other artifacts that are thought to be valuable enough to be preserved for the future. There is also the natural Cultural Heritage environment which incorporates components related to geology, paleontology, and morphology as well as landscapes, flora, and wildlife. Games can cover all the aforementioned material and therefore are an essential and integrated part of modern Cultural Heritage that can support teaching and learning for as a modern tool by integrating art, storytelling and digital technology [8,15].

### 2.2. VR as a Medium in Cultural Heritage

One of the main characteristics of VR is that headsets completely take over users' vision to give the impression that they are somewhere else. For games, VR supersedes the surroundings, taking the user to other places where physical presence is no longer important. These features provide numerous possibilities for VR games in cultural heritage, as the player is free to travel to different locations (real or fictional) across time and space and have experiences from different historical or even future eras. Dealing with VR, one inevitably deals with immersion, the feeling of being a part of the virtual environment. Although immersion is present in other mediums too, such as books and films, VR immersion is not only mental but could be also physical. In this case, the feeling of being physically present in the virtual world is strong, of course depending on the quality of the VR environment and its design [16].

However, the effective reconstruction of the virtual world for cultural heritage VR games is a challenge. Significant historical knowledge is needed in order to reconstruct past worlds in ways that are meaningful for the player and to allow her to play and explore history. Details on the appearance of monuments, objects, people, landscapes, etc., are very important in creating the right interaction conditions for the game [17]. In this sense, AR games for cultural heritage might be easier to implement than VR ones, as AR uses a real

world setting while VR needs to reconstruct entire worlds. In addition, AR might only need a smartphone for its use, whereas VR needs a headset device.

VR in cultural heritage is often combined with storytelling such that the effects of both VR and storytelling are maximized. In this setting, the person is using multiple modalities to access information, as she experiences the place where the story is evolving in the virtual world [18]. For example, iMareCulture explored ways storytelling, VR games, and underwater archaeology could be combined, as players could explore cities long lost under the sea (while playing in VR) [19].

Moreover, VR applications for cultural heritage can take many forms. Apart from games and reconstructions of different spaces, VR can be used in different and even unexpected forms of culture such as poetry [20], and researchers appear to be testing the medium and exploring its full potential. VR can also be used for testing research hypotheses, like assisting archaeological research. For example, VR and 3D models were used to explore architectural history and features of buildings that no longer exist [21].

For the development of virtual worlds and games for Cultural Heritage, different techniques are used. In the case of the historic Klodzko Fortress in Poland's Lower Silesia, terrestrial laser scanning was used to capture the real world. The scans were later used in a game engine to create virtual interactions [22]. The use of game engines for the implementation of virtual cultural experiences and VR culture games is popular with different researchers, since they provide a solution in reducing the production cost of VR applications and also in helping to overcome different technical issues [23].

Gamification of VR experiences is a topic of research that has been particularly increased with the growth of VR technology, such as, low-cost HMDs and high field-of-view. In [24], researchers compared different human–computer interaction methodologies for real-time VR simulation of both tangible and intangible digital heritage sites and the creation of dedicated, immersive, gamified curation experiences. Moreover, various interaction methods, such as sensor-based, device-based, tangible, collaborative, multimodal, and hybrid interaction methods, have also been employed by immersive reality technologies to enable interaction with the virtual environments. Previous studies compare the existing VR technologies and interaction methods against their potential to enhance cultural learning in Virtual Heritage applications [25]. This study provides guidelines for the utilization of immersive technologies and interaction methods that can assist professionals in cultural heritage to predetermine their relevance to attain the intended objectives of an applications: "*(1) establish a contextual relationship between users, virtual content, and cultural context, (2) allow collaboration between users, and (3) enable engagement with the cultural context in the virtual environments and the virtual environment itself*" [25].

Finally, new methods of interacting with VR games are developed. Researchers investigated psychological responses to playing videogames using a VR HMD [26] and also investigated how cybersickness impacts the sense of presence one feels in the virtual environment, as well as how cybersickness affects enjoyment. The study reports that better technology does not affect the frequency or severity of cybersickness for players; but sensory conflict has a significant impact on how sick users become.

### 2.3. Learning Experiences with VR Games

It is nowadays widely accepted that games can significantly boost learning in cultural heritage, since they increase engagement and motivation [27]. Studies of VR games have focused on different features that can make them more learning effective for different audiences, such as different genders [27]. VR can turn passive visitors into active learners. VR does not remove importance from the real objects but it can improve the understanding of them, since it allows users to manipulate objects and be actively involved [28]. The importance of games and gamification in cultural heritage is further explored by [10], where current trends are analyzed.

Learning is not only an individual activity but also a social one. In fact, learning benefits are maximized when people interact in a social setting and cultural heritage sites

are important cultural and also social spaces. A recent study showed that people using VR from home could interact with others physically present at the museum that used AR, and together they could play and co-explore cultural content. The important findings of this work were used to inform the Mixed Reality Museum Co-Visit Theory that fosters collaboration in cultural heritage setting and amplifies learning outcomes [29]. Similarly, Dolezal et al. [30] created collaborative educational applications in immersive virtual environments that allowed people geography (explanation of "people geography"?).

Efforts have been made to make the development of educational VR games for cultural heritage more effective. In particular, Game Engine Platforms are used to create Virtual Learning Environments as a response to the increased need for VR games and experiences due to the COVID-19 pandemic and restriction of physical visits [31].

## 3. Methodology

This SLR seeks to study VR games in Cultural Heritage settings. The research methodology adopted is based on the commonly accepted recommendations by [32,33]. PRISMA guidelines were followed for the best possible search outcomes, which entails establishing the needs of the research and the review process, automatic searches in well-known databases and evaluation of the studies, followed by data extraction.

### 3.1. Review-Research Questions

The research questions (RQ) of this SLR, focus on the outcomes of previous works. The following RQs highlight available data, gaps, and potential future directions of VR game in cultural heritage:

RQ1: How well-suited are VR games as a method or tool for cultural heritage?

*Analysis*: Focus on positive or negative impacts and effects of VR games in cultural heritage settings.

RQ2: How can we use VR games to support cultural heritage?

*Analysis*: Focus on the elements with which the players interact in a VR game and show effects for research and practice (e.g., usage, learning gains).

### 3.2. Search Strategy

In order to ensure that the analysis addressed most of the associated research publications, generic phrases were first utilized [33]. The key search terms were "Cultural Heritage" and "Virtual Reality." We followed the process outlined in Table 1 to complete our search phrase:

- Discovered main terms by identifying the main concepts from the RQs.
- Identified alternative synonyms (or spellings) for these terms.
- Checked the keywords in relevant studies.
- Added alternatives spellings and synonyms by using Boolean OR.
- Linked main terms by using Boolean AND.

We looked through the following online bibliographic databases to find high-quality information: IEEE Xplore, ACM Digital Library, Science Direct, Springer Link, and ERIC. In addition, we conducted independent research in considerable academic articles and conferences, e.g., IEEE Conference on Virtual Reality and 3D User Interfaces (VR).

**Table 1.** Databases and search process followed in this study.

| Database | Search Terms | Search Criteria |
|---|---|---|
| IEEE Xplore | ("Abstract":game OR "Abstract":videogame) AND ("Abstract":vr OR "Abstract":"virtual reality") AND ("Abstract":culture OR "Abstract":cultural OR "Abstract":heritage OR "Abstract":museum OR "Abstract":exhibition) | field Abstract; all publication dates; 77 initial result; |
| ACM DL | [[Abstract: game] OR [Abstract: videogame]] AND [[Abstract: culture] OR [Abstract: cultural] OR [Abstract: heritage] OR [Abstract: museum] OR [Abstract: exhibition]] AND [[Abstract: vr] OR [Abstract: "virtual reality"]] | ACM Full-Text Collection; field Abstract; all publication dates; 59 initial results; |
| ScienceDirect | ("virtual reality" OR VR) AND (game OR videogame) AND (culture OR cultural OR heritage OR museum OR exhibition) | fields: title, abstract or author-specified keywords; all publication dates; 28 initial results; |
| Springer Link | ("virtual reality" OR VR) NEAR (game OR videogame) NEAR (culture OR cultural OR heritage) | all fields; search for articles, books, chapters, conference papers; all publication dates; 82 initial results; |
| ERIC | ("virtual reality" OR VR) AND (game OR videogame) AND (culture OR cultural OR heritage OR museum OR exhibition) | fields: title, author, source, abstract and descriptor; all publication dates; 22 initial results; |

### 3.3. Inclusion and Exclusion Criteria

The data extraction process followed 3 rounds: (1) A prime review was conducted in order to gather data centered on common information and inclusion/exclusion criteria. The search results were carefully examined based on the paper's title, abstract, keywords, and conclusions. (2) Subsequently, each publication underwent a more thorough assessment in order to compile fundamental information about VR games as well as the level of the study issue and research topic. (3) Lastly, in the third round, papers were evaluated based on the criteria as shown in Table 2 and their positive contribution to cultural heritage through a full-text analysis.

**Table 2.** Primary works—Inclusion and exclusion criteria.

| Type | Criteria |
|---|---|
| Basic aspects | Title, Authors, Publication year, Article Type, Journal or Conference name, Publisher, Number of Citations, Extraction date |
| Inclusion & Exclusion Criteria | Activities irrelevant with VR games<br>Combine VR games *AND* Cultural Heritage<br>Empirical research *OR* VR gaming technologies<br>Well-structured research<br>Inadequate evaluation methods *OR* unclear findings *OR* insufficient data<br>Available online<br>English Language<br>Peer-reviewed |
| Review Questions<br>Research's purpose | Extent to which RQ1 and RQ2 are addressed<br>Use of VR games in cultural heritage, benefits or drawbacks reported, challenges reported |

*3.4. Quality Assessment*

Using selected criteria, we followed a quality assessment process to determine the consistency of the papers found [32]. In order to assess the quality of the selected work the following were considered: (1) Design of the studies and their effectiveness. (2) Methodology in terms of participants, tools, data measurement, and analysis. (3) Consistency between studies' results analysis and conclusions. (4) Study's findings applicable to other stakeholders (such as researchers and game developers). (5) Selected studies in correlation with the RQs of the present SLR.

Each paper was graded on a scale of 1 to 3 (low, moderate, and high) by two academics who worked separately on the evaluation. When the results varied by two points, the procedure was repeated by a third researcher, and the top two results were ultimately used. The cumulative weight of proof (woe) for each publication was taken into consideration by applying scores on each of the five variables. The scale went from 5 to 15 (low 5 and high 15). This process has been used in previous work and was found effective in assessing studies for review purposes [34].

The histogram in Figure 1 shows the apparent variations in the chosen threshold ratings. Most of the chosen research attained a high score since they were published in peer-reviewed scientific journals and conferences. The average position was 10.93 (it was expected to be high).

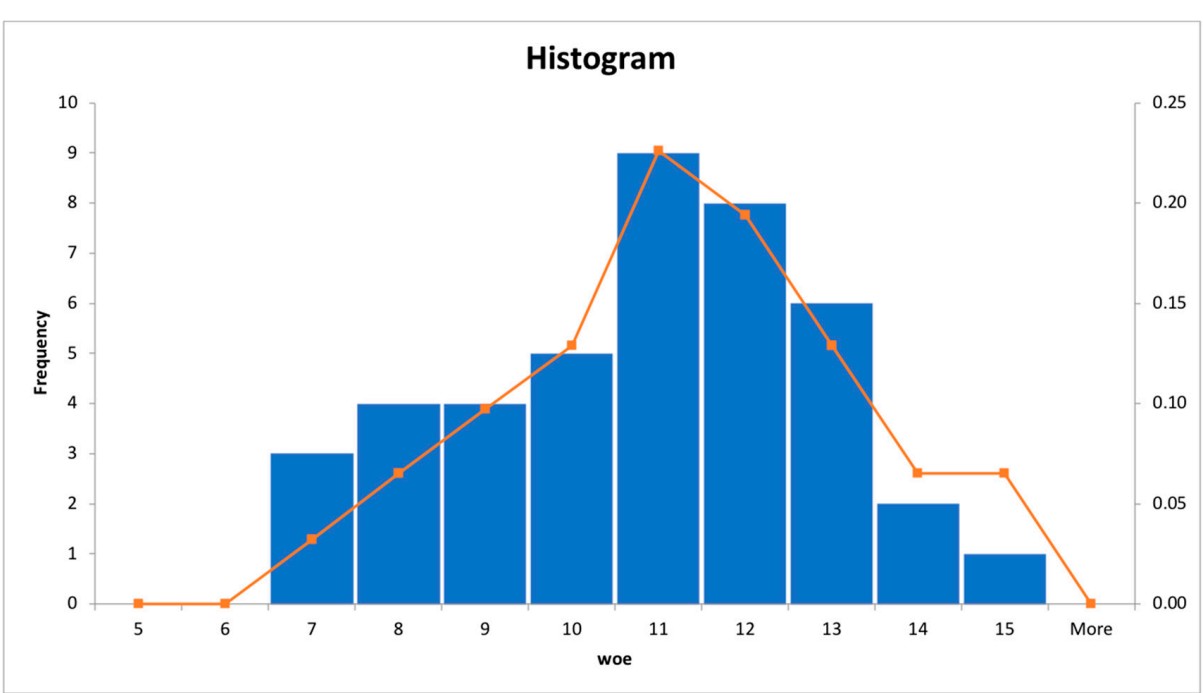

**Figure 1.** Quality assessment of included studies (histogram).

*3.5. Coding Scheme*

Initially, basic information was extracted from the reviewed papers, such as paper title, type of publication, publication year, and publisher. Furthermore, based on RQs of this SLR, different categories were made to reflect the use of VR games in the cultural heritage context, as shown in Table 3.

**Table 3.** Data extraction—coding scheme.

| Feature | Explanation |
| --- | --- |
| Research Purpose | indicates the purpose of the research in using VR games in cultural heritage |
| Research Design | identifies the method used to evaluate the VR gaming intervention |
| Participants | reports the participants using VR games |
| VR Technology | indicates technologies and tools used in the design and implementation of a VR game |
| Application Type | indicates the user interaction with the VR gaming environment and immersion |
| Learning effects | identifies learning effects observed or reported related cultural heritage |
| Game characteristics | identifies game design concepts and gaming characteristics |

## 4. Results

265 publications that matched the search terms were discovered after the search procedure (duplicate records removed). After looking through the title, abstract, and keywords, the following step was to apply filters (screening). The inter-rater reliability then showed that there was substantial agreement in the fields of title ($\kappa$ = 55.3%) and abstract ($\kappa$ = 66.2%), leading to the final selection of 59 works. The 59 publications were reviewed based on their full texts during the subsequent phase (eligibility) of the filtering process, leaving 46 studies. Then, after applying the exclusion criteria (shown in Figure 2), 41 papers were approved for the data extraction stage, leaving 5 more works behind. The snowball method was then used to add 1 more publication, bringing the total number of studies chosen for this systematic review to 42 (Table 4). The selection and filtering process is presented following the PRISMA guidelines, in Figure 2.

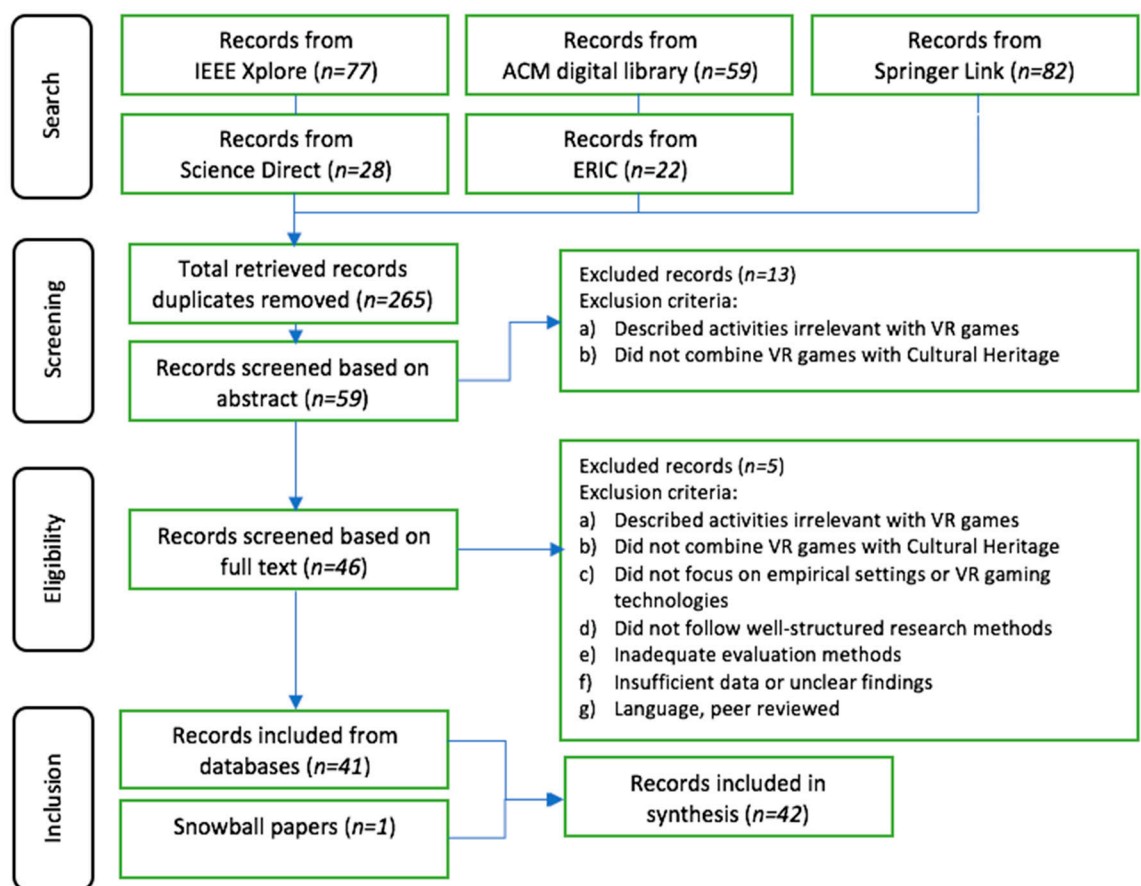

**Figure 2.** Flow of information through the different phases of this review (Prisma flowchart [33]).

**Table 4.** Selected studies in this review.

| ID | Title—Research Purpose—VR Gaming Experience | Reference |
|----|---------------------------------------------|-----------|
| S01 | Development and integration of digital technologies addressed to raise awareness and access to European underwater cultural heritage. An overview of the H2020 i-MARECULTURE project | [35] |
| S02 | Mixed Reality for Museum Experiences: A Co-Creative Tactile-immersive Virtual Coloring Serious Game | [36] |
| S03 | Application of Brain-Computer Interface and Virtual Reality in Advancing Cultural Experience | [37] |
| S04 | Knowledge Analysis Automatic Evaluation in Virtual Reality Immersive Experiences | [38] |
| S05 | From Spiritual to Virtual: An Interactive Digital Art Creation of Virtual Reality Borobudur | [39] |
| S06 | Go your own way: User preference in a time-based virtual heritage world | [40] |
| S07 | Predicting tourists' behavior of virtual museum using support vector machine with feature selection technique | [41] |
| S08 | [DC] Immersive Gamified Environments (IGE) as an Approach to Assess Subjective Qualities of Daylighting in Architectural Spaces | [42] |
| S09 | New Media and Space: An Empirical Study of Learning and Enjoyment Through Museum Hybrid Space | [43] |
| S10 | User experience evaluation of virtual reality-based cultural gamification using GameFlow approach | [44] |
| S11 | Recreating Little Manila through a Virtual Reality Serious Game | [45] |
| S12 | A fulldome interactive visitor experience a novel approach to delivering interactive virtual heritage experiences to group audiences in fulldome projection spaces, evaluated through spatial awareness and emotional response | [46] |
| S13 | Using Virtual Environments to Tell the Story: "The Battle of Thermopylae" | [47] |
| S14 | Application of Mixed Reality Technology in Education with the case of a Huangmei Opera Cultural Education System | [48] |
| S15 | MediaEvo project: A serious game for the edutainment | [49] |
| S16 | A simulation of life in a medieval town for edutainment and touristic promotion | [50] |
| S17 | Virtual Reality meets Degas: an immersive framework for art exploration and learning | [51] |
| S18 | Experiencing a town of the Middle Ages: An application for the edutainment in cultural heritage | [52] |
| S19 | Virtual reality technology based developmental designs of multiplayer-interaction-supporting exhibits of science museums: taking the exhibit of "virtual experience on an aircraft carrier" in China science and technology museum as an example | [53] |
| S20 | Design and Implementation of "Winning Luding Bridge" Immersion FPS Game Based on Unity3D Technology | [54] |
| S21 | Conceptualizing Embodied Pedagogical Mediation (EPM): The Plávana Project, A Choreographer's Toolkit. | [55] |
| S22 | Virtual Participation in Ukiyo-e Appreciation using Body Motion | [56] |
| S23 | RelicVR: A Virtual Reality Game for Active Exploration of Archaeological Relics | [57] |
| S24 | Research on the Application of VR Animation Technology in Traditional Folk Game Demonstration——: Take the traditional game pyramid in Dunhuang fresco as an example | [58] |
| S25 | "Meet the Deer King": "Splash-Ink" Interaction in the Innovative VR Game Based on Dunhuang Art and Culture | [59] |
| S26 | Generating Embodied Storytelling and Interactive Experience of China Intangible Cultural Heritage "Hua'er" in Virtual Reality | [60] |
| S27 | 3D reconstruction and validation of historical background for immersive VR applications and games: The case study of the Forum of Augustus in Rome | [61] |
| S28 | An integrated VR/AR framework for user-centric interactive experience of cultural heritage: The ArkaeVision project | [62] |
| S29 | Principles for the Design of a History and Heritage Game Based on the Evaluation of Immersive Virtual Reality Video Games | [63] |

**Table 4.** *Cont.*

| ID | Title—Research Purpose—VR Gaming Experience | Reference |
|---|---|---|
| S30 | Museum beyond physical walls: an exploration of virtual reality-enhanced experience in an exhibition-like space | [64] |
| S31 | Virtual Reality Arcade Game in Game-Based Learning for Cultural Heritage | [65] |
| S32 | VR Games and the Dissemination of Cultural Heritage | [66] |
| S33 | Creative Industries and Immersive Technologies for Training, Understanding and Communication in Cultural Heritage | [67] |
| S34 | An Investigation of Dissemination and Retention of Non-verbal Information About the Cultural Heritage of Rock Art in a Virtual Reality Simulation | [68] |
| S35 | Immersivity and Playability Evaluation of a Game Experience in Cultural Heritage | [69] |
| S36 | Puzzle Battle 2.0: A Revisited Serious Game in VR During Pandemic's Period | [70] |
| S37 | Dissemination of São Tomé and Príncipe Culture Through Virtual Reality: Comparative UX Study Between Potential Tourists from Portugal and Santomean Inhabitants | [71] |
| S38 | Optimization of Cultural Heritage Virtual Environments for Gaming Applications | [72] |
| S39 | The Role of Second Life Games in Promoting Cultural Heritage | [73] |
| S40 | Playhist: Play and Learn History. Learning with a Historical Game vs. An Interactive Film | [74] |
| S41 | Tourism and Virtual Reality: User Experience Evaluation of a Virtual Environment Prototype | [75] |
| S42 | Fantasy Gaming and Virtual Heritage | [76] |

Only $n = 5$ (12%) of the papers were published in peer-reviewed journals, whereas $n = 37$ (88%) were published in conference proceedings. They are from various fields of research, such as VR (e.g., S02, S08, S30), games and player interaction (e.g., S13, S21, S23), graphics and visualization (e.g., S09, S19, S25), and digital application in heritage and archaeology (e.g., S02, S11, S12, S27, S28, S33). Figure 3 presents the detailed list of publication venues where IEEE Digital HERITAGE ($n = 3$) and Springer EuroMed ($n = 3$) conferences comprised those with the highest number of contributions.

The analysis also shows that there was a large increase in number of publications ($n = 39$) during 2018 and 2022 (Figure 4), while the first paper was written in 2011. The distribution shows a growing interest in using VR games in cultural heritage.

Most studies ($n = 35$) used an experimental setting involving VR games. A small number of studies ($n = 7$) presented game scenarios and envisioned future uses of VR games in cultural heritage and did not report any experimental data.

### 4.1. VR Games—Charactersitics

This section identifies gaming concepts and characteristics and indicates technologies and tools used in the design and implementation of VR games in this review.

VR technology can give a truly immersive, first-person perspective of game action. VR games can be played on standalone systems, specialized game consoles, or using advanced structures. Participants in VR games may experience and alter the cultural environment through a variety of devices and accessories, including headsets, VR gloves, and hand controllers with sensors, etc. (Table 5). Actions in the game are produced by the player's input, which includes head, hand, and body movement as well as any controller buttons or triggers. In VR gaming environments the player's perspective will typically track with how they move their hand, and they will be shown virtual hands to let them interact with the environment.

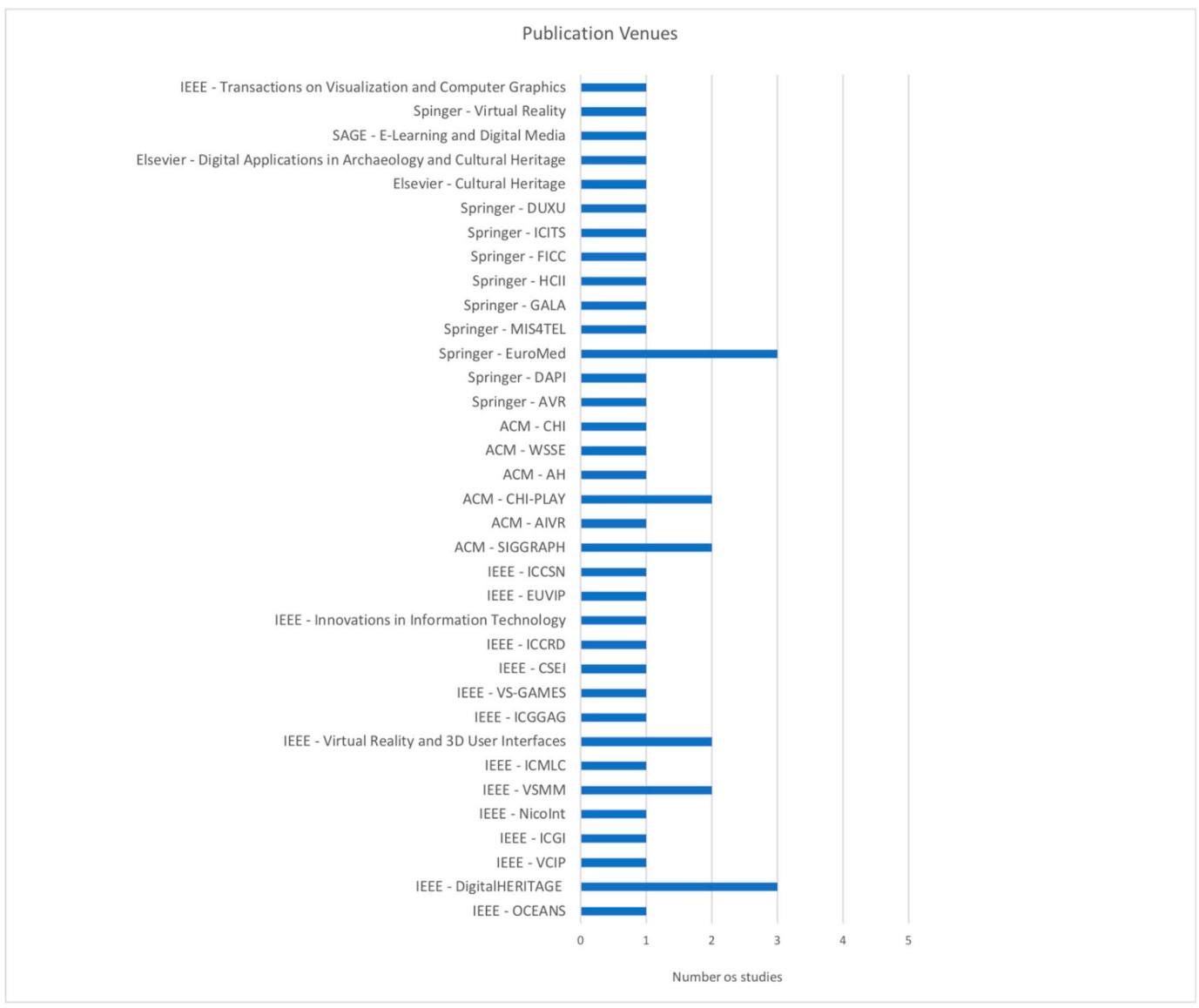

**Figure 3.** Publication venues of selected works.

All the papers analyzed serious games, as these are games with a purpose other than solely for entertainment. Table 6 shows different types of serious VR games. The domain of cultural heritage implies that while playing, the users will interact with cultural and/or historical content. In doing so, the users increase learning possibilities and therefore, games in cultural heritage are considered serious games. Serious games can use different methodologies to engage users and present historical and cultural phenomena, such as storytelling.

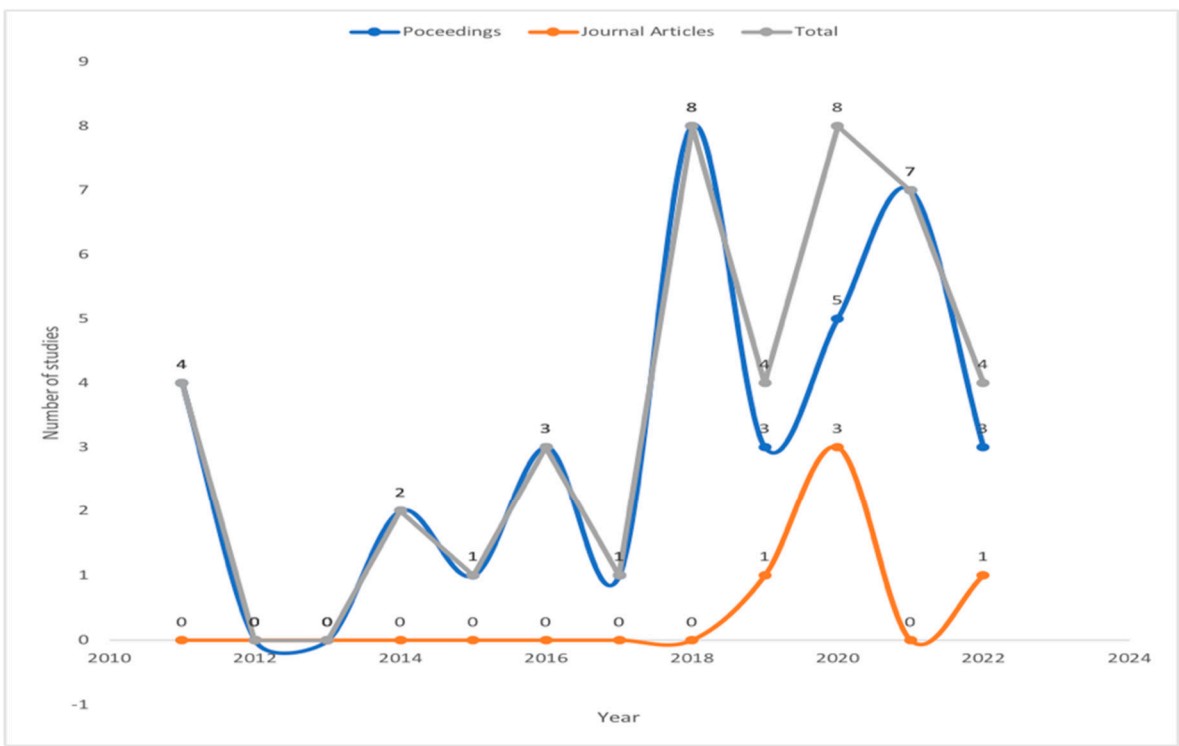

**Figure 4.** Distribution of selected papers per year.

**Table 5.** VR technology and/or functionality (selected studies).

| Tools—Functionality | References |
|---|---|
| HMDs | S01, S02, S03, S04, S05, S06, S08, S09, S11, S17, S19, S20, S21, S22, S23, S24, S25, S26, S28, S29, S30, S31, S32, S34, S35, S36, S37, S38, S39, S40, S41, S42 |
| Dome projection, caves, holographic | S01, S12, S14 |
| Cardboards-mobile usage, other | S07, S10, S13, S15, S16, S18 |
| Multimodal interfaces | S02, S03, S06, S33 |
| Hand or body tracking | S02, S19, S22, S25, S26 |
| Eye-tracking, EEG | S03, S04 |

**Table 6.** VR game types (selected studies).

| Genre—Type | References |
|---|---|
| Serious game | S01, S02, S03, S04, S05, S06, S07, S08, S09, S10, S11, S12, S13, S14, S15, S16, S17, S18, S19, S20, S21, S22, S23, S24, S25, S26, S27, S28, S29, S230, S31, S32, S33, S34, S35, S36, S37, S38, S39, S40, S41, S42 |
| Interactive storytelling | S01, S04, S05, S09, S11, S12, S13, S14, S15, S16, S18, S22, S24, S25, S26, S28, S29, S33, S35, S37 |
| Escape, puzzle, quest game | S03, S05, S06, S11, S23, S30, S31, S34, S36 |
| Beat 'em up game | S10, S20 |
| Co-op—multiplayer | S02, S12, S19, S26, S28 |

In interactive storytelling, in for example S01, S04, and S11, players take the role of immigrants unfamiliar with their surroundings, players engage in tasks that inform them what life was like in Little Manila for Filipino Americans in the mid-20th century, and are educated about the importance of Little Manila in American history. Another example of engaging people with history, is presented in S33 which shows the transformation of historical references into a meaningful three-dimensional video animation production and the design of a full-immersive serious game in VR.

The power of VR games of immersing users in various real or unreal situations that could not be experienced in other ways are presented in S01, which allows underwater cultural experiences, and in S02, where users play with dinosaurs.

In other works, S06 explores the user experience of a multimodal, time-based, virtual environment, and S14 provides textual and graphic explanations of the Huangmei opera along with performances and cultural interactive self-learning processes that are realistically presented to the HoloLens user.

Researchers are also involved with assessing the effect of VR games for cultural heritage. For example, in S03 users' gaze direction and time spent on focusing on specific details was used to reveal attention.

There are also a few works that focus on social aspects of the VR experience. For example, in S19 five players are immersed into virtual scene via VR goggles, and every participant could see virtual avatars of four other participants in the virtual scene. In the virtual scene, these participants coordinate with each other, jointly completing the multiplayer game task to transport the carrier-based aircraft from the hangar to the takeoff location on the flight deck and realize takeoff. Interactivity is therefore central in VR games which can turn a passive visitor to an active one, increasing visitor motivation and popularity of the venue and its exhibits.

### 4.2. Cultural Practices—Learning Gains

This section identifies learning effects observed or reported from VR interventions with games used in cultural heritage (Table 7). VR games for cultural heritage can teach cultural awareness, historical reconstruction, and heritage awareness in a highly immersive and engaging manner, e.g., S11.

**Table 7.** Learning effects (selected studies).

| Effects | References |
| --- | --- |
| Raise awareness | S01, S03, S05, S11, S14, S20, S21, S22, S26, S31 |
| Learning gains | S01, S02, S03, S04, S05, S06, S09, S11, S13, S14, S15, S16, S17, S18, S21, S22, S23, S24, S26, S28, S29, S31, S32, S33, S34, S35, S37, S38, S40, S41, S42 |
| Enjoyment—engagement | S02, S03, S04, S05, S06, S09, S13, S14, S17, S19, S20, S21, S22, S24, S25, S26, S28, S30, S31, S32, S33, S36, S39, S42 |
| Personalization | S01, S05, S06, S08, S10, S11, S12, S14, S17, S37 |

VR technology has the potential to allow players to interact with the virtually recreated world, thus enhancing player experience in ways that other digital technologies cannot (S11). For example, studies show important learning gains from the use of such games (S09) and there are reports on learning of specific aspects of culture and tradition (S01, S03). VR games allow Cultural Heritage research to go one step further than recreation of virtual content and can actively facilitate cultural learning [77]. Learning benefits seem to be significant also when VR games manage to activate players' emotions, e.g., S30.

In S15, S16, and S18, the game is intended as a means to experience a reconstruction of everyday life and activities of the people during the age of Frederick (XIII century), describing life in simple language but preserving historical facts with utmost detail.

Moreover, users report joy when they play cultural heritage VR games (S09) and they particularly enjoyed the interactive nature of the experience (S37). They can also have the opportunity of a personalized experience (S01) and, depending on their personality, people seem to engage more frequently with such games (S06, S10). In S08, tourist behavior can be predicted from the gaming behavior of users.

There are also examples of VR games that present intangible heritage. For example, S24 presents the utilization of VR animation technology on intangible cultural heritage and traditional folk games through the traditional game pyramid in Dunhuang fresco.

In S26 Hua'er, a type of traditional oral performance is one of the national intangible types of cultural heritage (ICH) in China. An interactive VR system is used to engage audiences to experience and understand the connotation of Hua'er performance. S32 developed the VR Terracotta Warriors Serious Games which is part of a research series on Key Technologies of the Smart Museum for the Audience. The VR games combine the live experience of the audience, using 360° panorama shooting, 3D modeling, virtual reality and intelligent, question-answering technology to design these games. From the real scene to the virtual scene, the audience can experience a wonderful journey through time and space and learn the history and culture of the Qinshihuang's Mausoleum.

### 4.3. Player Experience

This section identifies aspects regarding the players' experience in VR gaming environments in terms of immersion and describes emotional responses evoked when the player interacts with the VR game system (Table 8).

**Table 8.** Player experience (selected studies).

| Effects | References |
|---|---|
| Immersion | S01, S02, S03, S04, S05, S06, S07, S08, S09, S10, S11, S12, S13, S14, S15, S16, S17, S18, S19, S20, S21, S22, S23, S24, S25, S26, S27, S28, S29, S230, S31, S32, S33, S34, S35, S36, S37, S38, S39, S40, S41, S42 |
| Usability | S01, S02, S03, S06, S07, S08, S09, S10, S12, S17, S19, S22, S23, S26, S27, S38, S41 |
| Aesthetics | S01, S02, S04, S05, S08, S09, S11, S12, S21, S22, S26, |
| Physically impossible experience | S01, S02, S06, S13, S15, S17, S18, S20, S22, S23, S26, S27, S28, S30, S32, S34, S35, S42 |
| Feedback | S03, S06, S22 |
| Training needed | S08, S12 |
| Motion sickness | S08, S09 |

User experience is crucial in VR games since they can directly influence the levels of immersion and enjoyment. User interaction in the VR gaming environment raises significant design challenges (e.g., S10) and demand specific solutions. Similarly, VR games need to be usable and respect usability guidelines. For instance, several locomotion techniques on player perception and system usability within the environment should be further investigated (S08).

Players also seem to like the fact that VR games allow them to experience worlds that are impossible to experience in the physical reality. They enjoy such experiences because certain locations are impossible to reach, because the events took place in the past, or even because the story is based on a fictional setting. (e.g., S01, S02). The increased interest in such games is recorded in different works that even report long queues of people waiting to experience the virtual world (S28).

VR games also apply aesthetic principles, not only to make the gaming environment attractive but also to recreate the historical details of monuments and objects, like in the case of the Plávana Project that allowed people to explore the Konark Temple Grounds (S21).

Many works focus on the evaluation of the gaming experiences. In particular, researchers are interested in the effects of such games on the cultural experience. From self-reports to physiological data coming from EEGs (i.e., S03), the user experience is analyzed as well as the learning gains. People seem able to connect the virtual content with the physical one and thus, their virtual experience seems to significantly enhance the physical one (S06).

S09 highlights the importance of the training of player-users, as there are many people who are not familiar with VR equipment. Most players are used to standard controllers for gaming consoles and computers and using a new kind of controller (e.g., VIVE controllers) might be obtrusive for many audiences.

Moreover, VR-enhanced museums should be able to serve multiple participants simultaneously (S09) and this might pose administrative challenges.

Finally, certain studies also report further problems with VR games in cultural heritage, such as motion sickness (S07, S09)

## 5. Discussion

This section discusses the abovementioned results and how they can be interpreted from the perspective of previous studies corresponding to the RQs of this review. The findings and their implications are discussed while future research directions and limitation of this study are also highlighted.

### 5.1. RQ1: Positive and Negative Impacts of VR Games

VR games and cultural heritage settings are in the process of converging, and their interaction can be mutually beneficial. VR games not only give players a glimpse of the past, but allow them to manipulate the virtual word, be active in it, interact with cultural content and other players and actively involve perception and imagination (S13). As presented above, the use of VR games in cultural heritage are associated with multiple learning gains, increased visit motivation, and dynamic engagement. The ability of VR to increase immersion allows people to experience culture with multiple senses and not only to hear or read about it.

Multiple types of technologies can be used depending on the venues needs and multiple types of games and interactions can be implemented, covering the needs of diverse audiences. These applications allow people to experience culture alone or in groups, and also provide opportunities for personalized content and experiences, depending on users' needs, personalities, and interests. VR games seem to increase enjoyment and thus provide a more holistic experience that not only focuses on learning but also addresses other needs of cultural visitors, such as having fun, being active, and making meaningful interactions with cultural content and other visitors. Finally, VR games can allow researchers and museum staff to collect valuable visitor feedback.

In the works reviewed, a few negative impacts were also reported. One is related to possible training needs of players before they can successfully use the virtual environment. However, a design that respects usability principles can lower such demands. Another is related to the motion sickens reported in some studies. To investigate motion sickness in VR games, different studies have used the Simulator Sickness Questionnaire (SSQ) as a measurement tool [78,79]. Usability principles together with specific guidelines for the use of such games (e.g., time restrictions, etc.) can also prevent such symptoms. Different genre of environments may provide significant differences in the levels of motion sickness [80].

### 5.2. RQ2: VR Games to Support Cultural Heritage

Concerning interaction in VR gaming environments, it is important that players have meaningful choices. The game structure and rules should allow players to have an effect

on events and influence the game outcomes. Players need to clearly understand how their actions change the outcomes and believe they have some control over the game. It is also important to find a balance between too much or too little ambiguity. The game should have surprise elements but not to the point where the player believes that she has no control. This implies that the elements of the game work in combination to create a cultural heritage environment that is appropriately challenging, but which players still perceive as fair and equitable, balancing fun and learning. Sometimes finding the right balance between a game that is too hard or too easy to play is difficult. In addition, such games need to find a balance between gaming elements and cultural content that respects historical and cultural data.

VR games in the field of cultural heritage can be used to increase its sustainability. When used within a specific sustainability strategy, the games can have maximum effect [81]. UX methodologies are now mature enough to be used in cultural technologies and VR games in order to provide smooth visitor experiences and to allow useful evaluation of visitor experiences [82]. In addition, the field of games in cultural heritage is expanding, with new types of games emerging, like cinematographic videogames [83]. It is important to see where VR games stand compared to other types of games for cultural heritage and explore what is cost effective, maximizes visit effects, increases learning and engagement, etc. Moreover, it is also important to consider the physical aspects of VR games, such as the equipment used, as well as where they will be played (e.g., at the museum, at people's homes) ([80], p. 8). Finally, it is crucial to consider organizational issues around the use of VR games in cultural heritage, such as possible organization resistance issues, acceptance levels, as well as the mode of use (i.e., whether we will isolate the individual in the virtual world or we will support virtual social interactions). As previous study reports, the spectatorship experience for VR games differs strongly from its non-VR precursor [84]. The immersive full-body interaction is a crucial part of the player experience. Bystanders and viewers prefer the first-person version, which allows them to better focus on in-game actions and experience greater involvement [84].

### 5.3. Limitations of This Systematic Review

There are a few limitations that should be taken into consideration. First, the present work is by no means exhaustive, and many other relevant works might exist. The works cited here were the product of specific search processes (i.e., specific keywords) and other results might have emerged if other search processes were followed. We also restricted our search to specific types of works, like papers from conference proceedings and journals, excluding others like posters, uploaded works in academic repositories, etc. Filtering was largely manual, meaning that human factors might have altered the search results and/or the interpretation of results. In order to minimize this bias, all the chosen papers were read many times by different individuals. Following the PRIMA guidelines, more than one researcher dealt with the papers and possible deviations were discussed by the group of researchers. Finally, the review questions for this study demand responses that are not binary, meaning that there are overlaps and room for various interpretations. Another problem we faced was the fact that terminology is sometimes used in different manners by different researchers and it becomes difficult to draw conclusions.

### 6. Conclusions

This systematic literature review offers an analysis of the current state of VR games and cultural heritage. VR appears to be a very promising technology in facilitating experiences in the field. It offers significant results concerning the types of technologies used, the types of games used, as well as the different types of experiences for the user. Usage of VR games in the cultural heritage area is associated with multiple learning gains, increased visit motivation, and dynamic engagement. Given the increasing interest of museology and VR games, we expect VR games in to continue expanding over the next years, as they can support onsite as well as offsite cultural experiences. This article may help readers deepen

their understanding of this novel yet crucial topic, and may influence the development of the next wave of VR games for cultural heritage. Future work will further investigate the specific ways in which VR games may support different aspects of the cultural experience and how those factors could be involved in the physical, temporal, social, and individual aspects of the visit.

**Author Contributions:** Conceptualization, A.T.; methodology, A.T.; software, A.T.; validation, A.T. and A.A.; formal analysis, A.T.; investigation, A.T.; resources, A.T. and A.A.; data curation, A.T.; writing—original draft preparation, A.T. and A.A.; writing—review and editing, A.A.; visualization, A.T.; supervision, A.T. and AA.; project administration, A.T. All authors have read and agreed to the published version of the manuscript.

**Funding:** This research received no external funding.

**Institutional Review Board Statement:** Not applicable.

**Informed Consent Statement:** Not applicable.

**Acknowledgments:** The authors would like to thank Thomas Lunde for proofreading the manuscript and providing comments and the anonymous reviewers for their valuable comments.

**Conflicts of Interest:** The authors declare no conflict of interest.

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
