# Peer review of "VR Games in Cultural Heritage: A Systematic Review of the Emerging Fields of Virtual Reality and Culture Games"

_applsci, doi:10.3390/app12178476_

Round 1

Reviewer 1 Report

(Comment 1)

In this manuscript, the authors try to analyze how VR games are used in Cultural Heritage settings. The manuscript shows the quantitative and qualitative findings of VR games based on an investigation of many related papers. In addition, this paper will contribute to various people such as researchers related to Cultural Heritage and engineers of VR technology.

From these contributions, I think that this paper is suitable for publication in Applied Sciences. In order to publish, minor revisions with further several explanations might be necessary.

(Comment 2)

I recommend explaining previous research or activities for learning  Cultural Heritage other than VR games. 

(Comment 3)

Could you mention the reasons for the selection of online bibliographic databases (IEEE Xplore, ACM Digital Library, Science Direct, Springer Link and ERIC) ?

Maybe, these databases could cover technical research about VR games; however, several databases such as IEEE Xplore might be far from  Cultural Heritage.

(Comment 4)

Please modify the resolution and size of the font in the images. Especially, the resolution of the images is poor.

(Comment 5)

I think that it is interesting few negative impacts of VR games obtained from your investigation. If there are further solutions for avoiding several issues such as motion sickness, please add explanations.

(Comment 6)

Are there comparisons for commercial current devices or interfaces for VR games? The advantages and disadvantages of each device will contribute to implementing  VR games.

(Comment 7)

In the "5. Conclusions", I cannot imagine the major results and overall conclusion of this review paper. In the conclusion part, the major results and overall conclusion should be mentioned in simple and short sentences.

Reviewer 2 Report

The authors present a systematic review of papers for games in cultural heritage based on virtual reality technologies.  

The structure of the paper is straightforward and follows a standard methodology for literature reviews.

There are some typos. Table 1 is defined two times. There is probably an extra or unfinished statement in line 187.

In my opinion, 3D visualization and interactive storytelling do not count as games. It would be beneficial to make some explanation of what computer games are. Furthermore, the clarification of gamification will help explain how static content can be transformed into a serious game.

I am surprised by the low number of papers acquired and that only five of them are from peer-reviewed sources.
